# Effects of Si/Al Ratio on Passive NO_x_ Adsorption Performance over Pd/Beta Zeolites

**DOI:** 10.3390/molecules28083501

**Published:** 2023-04-16

**Authors:** Shasha Huang, Qiang Wang, Yulong Shan, Xiaoyan Shi, Zhongqi Liu, Hong He

**Affiliations:** 1Beijing Key Laboratory for Source Control Technology of Water Pollution, College of Environmental Science and Engineering, Beijing Forestry University, Beijing 100083, China; 2Engineering Research Center for Water Pollution Source Control & Eco-Remediation, College of Environmental Science and Engineering, Beijing Forestry University, Beijing 100083, China; 3State Key Joint Laboratory of Environment Simulation and Pollution Control, Research Center for Eco-Environmental Sciences, Chinese Academy of Sciences, Beijing 100085, Chinazqliu_st@rcees.ac.cn (Z.L.);; 4University of Chinese Academy of Sciences, Beijing 100049, China; 5Center for Excellence in Regional Atmospheric Environment, Institute of Urban Environment, Chinese Academy of Sciences, Xiamen 361021, China

**Keywords:** diesel exhaust, passive NO_x_ adsorber, Pd, Beta, Si/Al ratio

## Abstract

In the current article, the effect of Si/Al ratio on the NO_x_ adsorption and storage capacity over Pd/Beta with 1 wt% Pd loading was investigated. The XRD, ^27^Al NMR and ^29^Si NMR measurements were used to determine the structure of Pd/Beta zeolites. XAFS, XPS, CO-DRIFT, TEM and H_2_-TPR were used to identify the Pd species. The results showed that the NO_x_ adsorption and storage capacity on Pd/Beta zeolites gradually decreased with the increase of Si/Al ratio. Pd/Beta-Si (Si-rich, Si/Al~260) rarely has NO_x_ adsorption and storage capacity, while Pd/Beta-Al (Al-rich, Si/Al~6) and Pd/Beta-C (Common, Si/Al~25) exhibit excellent NO_x_ adsorption and storage capacity and suitable desorption temperature. Pd/Beta-C has slightly lower desorption temperature compared to Pd/Beta-Al. The NO_x_ adsorption and storage capacity increased for Pd/Beta-Al and Pd/Beta-C by hydrothermal aging treatment, while the NO_x_ adsorption and storage capacity on Pd/Beta-Si had no change.

## 1. Introduction

NO_x_ is an important precursor for ozone and haze pollution, which is harmful to natural environment and human health [1,2]. Most of the NO_x_ in the atmosphere comes from mobile sources, so it is very important to control the NO_x_ emission of diesel exhaust. At present, the most widely used and commercialized technology for removing NO_x_ from diesel vehicle exhaust is ammonia selective catalytic reduction (NH_3_-SCR) technology. However, NH_3_-SCR technology only has excellent removal efficiency of NO_x_ at high temperatures above 200 °C, while the NO_x_ reduction efficiency is extremely low when the exhaust temperature is below 200 °C [3,4,5]. With the increasingly stringent emissions regulations in future, controlling NO_x_ emissions from diesel vehicle exhaust during cold-start period is a considerable challenge due to the low temperature of the exhaust [6]. Therefore, the passive NO_x_ adsorber (PNA) technology came into being. The PNA technology can trap NO_x_ at low temperatures and release it at 200–500 °C, which is the temperature window of NH_3_-SCR catalyst working efficiently [7,8,9]. The desorbed NO_x_ is further removed by the downstream SCR catalyst, thus achieving the purpose of efficiently removing NO_x_ at low temperatures.

The most widely used PNA materials can be divided into two types. One is Pd dispersed on oxides (CeO_2_, Al_2_O_3_), but the weak hydrothermal stability and sulfur resistance limit their application [9,10,11,12,13]. In the past years, Pd/zeolites with various framework structures were paid much attention as PNA materials due to the large NO_x_ adsorption and storage capacity, appropriate desorption temperature and better resistance to hydrothermal aging and sulfur resistance [9,14,15,16,17,18].

The pore size and structure have significant influences on the PNA performance and stability of Pd/zeolites. Chen et al. investigated the PNA performance of Pd/BEA, Pd/MFI, Pd/CHA and Pd/CeO_2_ materials and found that Pd/BEA zeolites showed the largest NO_x_ storage capacity regardless of sulphation [9]. Further, Khivantsev et al. established the structure-storage property relationships of Pd/BEA zeolites as PNA materials [19]. They uncovered the inhibition effect of H_2_O and promotion effect of CO on PNA storage of Pd/BEA zeolites due to the formation of Pd(II)(NO)(CO) species. Pace et al. compared the distribution of Pd species in Pd/Beta and Pd/CHA zeolites under various pretreatment conditions and found that ionic Pd^2+^ in Pd/Beta is less stable and easy to form either Pd metal or PdO particles [20].

The Si/Al ratios of zeolite significantly influence the metal active sites and framework structure. Many researchers have proved that Pd^2+^ is the active site to store NO_x_ in Pd/zeolite that is mainly riveted on the framework Al. Therefore, the Si/Al ratio of zeolite has a significant impact on dispersion of Pd^2+^ [21]. The Al-rich zeolite favors the presence of highly dispersed Pd ions compared to the zeolite with high Si/Al ratio [22]. However, the structure of Al-rich zeolite is less stable. Zhao et al. reported that Pd/SSZ-13 with Si/Al ratio of 6 is easier to dealuminate at high temperatures than Pd/SSZ-13 with Si/Al ratio of 13, which damages the zeolite structure and leads to lower PNA activity [21]. In addition, the Si/Al ratio can also influence the NO_x_ desorption property of PNA materials. Mihai et al. found that Pd/Beta with a SiO_2_/Al_2_O_3_ ratio of 38 and 300 have two NO_x_ desorption peaks in the absence of CO at 100 °C and 180 °C, which is relatively low for NH_3_-SCR catalyst [23]. Nevertheless, Pd/Beta with a low SiO_2_/Al_2_O_3_ ratio of 25 have an additional desorption peak, despite only a small amount of NO_x_ release, in the absence of CO at 250 °C, where the downstream SCR catalyst can significantly decrease the NO_x_ [23]. In the presence of CO, the Pd/BEA with a SiO_2_/Al_2_O_3_ ratio of 25 and 38 showed an intensified NO_x_ desorption peak at 250–350 °C, while the Pd/BEA with a SiO_2_/Al_2_O_3_ ratio of 300 only showed a NO_x_ desorption peak at 100–200 °C. 

To sum up, Pd/BEA zeolite is a great potential candidate as PNA material due to its large NO_x_ adsorption and suitable desorption temperature as well as its economy and accessibility [9,20,24]. However, the Pd^2+^ species on Pd/BEA zeolites were less stable, resulting in weak hydrothermal stability, especially compared to the Pd/CHA zeolite with a small pore structure. Therefore, this work mainly concentrated on the framework and Pd speciation stability of Pd/Beta zeolites. The Pd-loaded Al-rich, common, and Si-rich Beta zeolites with Si/Al ratios of 6, 25 and 260 were selected as PNA materials before and after the hydrothermal aging treatment. The adsorption and storage amounts, desorption temperature and Pd species and framework structure were systematically investigated.

## 2. Results and Discussion

### 2.1. NO_x_ Adsorption/Storage Performance on Pd/Beta Zeolites with Various Si/Al Ratios

Figure 1 depicts the NO_x_ adsorption and storage profiles of Pd/Beta zeolites with various Si/Al ratios. The NO_x_ adsorption/storage amounts and absolute content of NO_x_/Pd (the ratio of the NO_x_ storage amount and Pd loading) were listed in Table 1. The Pd/Beta-Al zeolite shows a rapid adsorption capacity in the first 3 min and levels off at 20 min. The NO_x_ storage amount of Pd/Beta-Al zeolite is up to 65.0 umol/g. In the desorption temperature range, two peaks at 250 °C and 370 °C were observed, which located in the ideal operation window of Cu-zeolite NH_3_-SCR catalysts [4,25,26,27]. NO was the main desorption species, while the content of NO_2_ was lower (Appendix A). Moreover, the Pd/Beta-Al showed NO_x_/Pd storage of 0.69. This indicated that 69% of Pd species exist as isolated Pd^2+^, corroborating reports that dispersed Pd^2+^ ion species are active sites for NO_x_ adsorption [20]. The other two samples, however, only showed that 27% and 3% of Pd species are isolated Pd^2+^, indicating the severe accumulation of Pd species.

After hydrothermal aging treatment, the NO_x_ adsorption and desorption amount both increased clearly for Pd/Beta-Al. In addition, the adsorption time to saturation of hydrothermally aged samples is similar to the fresh ones, indicating an increasing adsorption rate of NO_x_. The NO_x_/Pd increased from 0.69 to 0.92 and 0.96 (Table 1) after hydrothermal aging treatment at 750 °C and 800 °C, respectively. This demonstrated that Pd species redispersed into ionic Pd^2+^ during hydrothermal aging treatment. Moreover, the desorption peak at 250 °C significantly intensified, accompanying the decrease of the desorption peak at 370 °C, again indicating the redistribution of Pd species. This is a beneficial change for diesel aftertreatment system, since 250 °C is sufficiently high for the SCR catalyst to efficiently eliminate NO_x_. Moreover, the Pd/Beta-Al can be easily recovered and adsorb NO_x_ at low temperatures again.

The Si/Al ratio has a significant impact on the activity and hydrothermal stability of Pd/Beta zeolites as PNA materials. The NO_x_ adsorption and storage capacity on Pd/Beta zeolites gradually decreased with the increase of Si/Al ratio. The desorption temperature also shifted to high temperatures with the increase of the Si/Al ratio. The Pd/Beta-C with a medium Si/Al ratio of 25 showed a lower adsorption ability and similar variation tendency after hydrothermal aging treatment compared to Pd/Beta-Al materials, with the result shown in Figure 1b. The NO_x_/Pd increased from 0.27 to 0.75 and 0.82 after hydrothermal aging at 750 °C and 800 °C treatment, respectively, which indicated that the redistribution of Pd species can also occur on the Pd/Beta-C zeolites. On the other hand, too high of a Si/Al ratio of Pd/Beta led to the deactivation of PNA performance regardless of hydrothermal aging, with the result shown in Figure 1c. Only a slight desorption peak of NO_x_ was observed at 290 °C for fresh Pd/Beta-Si zeolite. Next, the zeolite framework structure and Pd active species were analyzed.

### 2.2. The Structure Analysis of Pd/Beta Zeolites

To understand the crystallinity and structure of the zeolite, XRD characterization was carried out as shown in Figure 2. For the fresh Pd/Beta-Al zeolite, the diffraction peaks at 7.7°, 13.4°, 21.5°, 22.5°, 25.4°, 26.8°, 29.6° and 43.5° were corresponding to pure Beta phase crystallization [28,29,30]. Among them, the diffraction peaks at 13.4°, 22.5° and 25.4° were ascribed to (004), (302) and (304) crystal planes, respectively [31]. The weak signal at 22.5–32.5° appeared, which may be due to amorphous phase (non-framework Al species) [9]. The Pd/Beta zeolites with different Si/Al ratio have excellent crystallinity, which shows that Si/Al ratio has little effect on the crystallinity. After hydrothermal aging at 750 °C and 800 °C treatment, the crystallinity of Beta zeolites with different Si/Al ratios scarcely changed, indicating the high stability of BEA zeolite framework structure. For the identification of Pd species, diffraction peaks at 33.8° and 40.2° were observed, corresponding to the (101) crystal plane of PdO and (111) crystal plane of metal Pd, respectively [32,33,34]. For fresh and hydrothermally aged Pd/Beta-Al zeolites, no characteristic diffraction peaks of PdO and Pd species are observed, indicating the high disperse of Pd species regardless of hydrothermal aging treatment. For the Pd/Beta-C zeolite, only a slight diffraction peak of PdO was observed, and the peak intensity decreased after hydrothermal aging treatment, indicating redisperse of Pd species. However, besides the observation of the PdO diffraction peak, the diffraction peak, attributed to metal Pd, also appeared for the hydrothermally aged Pd/Beta-Si zeolites. This indicated Pd species were more facile to accumulate on the Si-rich Beta zeolite. Moreover, with progressively hydrothermal aging, the peak of metal Pd intensified accompanying the decrease of the peak intensity that represented the PdO species. This demonstrated that the accumulated PdO species turned into metal Pd rather than Pd^2+^ species due to the few ion exchange sites in the Si-rich Beta zeolite. 

In addition to XRD characterization of the zeolite framework, NMR was also used to characterize the local framework Si and Al atoms of zeolite. Figure 3 shows the ^29^Si NMR profiles of Pd/Beta zeolites, and the deconvolution diagram of ^29^Si NMR is shown in Appendix A. The peaks with chemical shifts at −115 ppm, −112 ppm and −110 ppm were assigned to Si (4Si) species [35,36,37]. The peak with chemical shifts at −103 ppm was assigned to Si (3Si, 1Al) or Si (3Si, 1OH) species [35,36,37]. Obviously, dealumination occurred and intensified with progressively hydrothermal aging for Pd/Beta-Al as indicated by the decrease of the peak at −103 ppm. Nevertheless, it should be noted that the hydrothermal aging treatment did not lead to deterioration in PNA activity, which may be attributed to the fact that the strong interaction between the framework and Pd ions’ coordinated Al sites can survive from dealumination [7]. The similar results are also observed in the Pd/Beta-C zeolite but the dealumination is weaker. For the Pd/Beta-Si zeolite, however, there only weak dealumination occurs due to the few framework Al sites. The results of ^27^Al NMR measurements were consistent with ^29^Si NMR results as shown in Figure 4. Pd/Beta with different Si/Al ratios show resonances at chemical shifts of 0 ppm and 54 ppm, which are attributed to the octahedral extra-framework Al and tetrahedral Al in the framework, respectively [38,39]. An additional peak was also observed, which was attributed to pentahedral Al [19]. With the decrease of Si/Al ratio, dealumination occurred more easily during hydrothermal aging. For example, 40% of framework Al was reserved for Pd/Beta-Al, while 70% of framework Al was reserved for the Pd/Beta-C zeolite, and there is no obvious change of framework Al peak for the Pd/Beta-Si zeolite after hydrothermal aging at 800 °C treatment. Attentionally, some framework Al suffered from hydrothermal aging and favored the Pd distribution, which may be the reason that the Pd/Beta-Al zeolite behaved the best in PNA performance and hydrothermal stability despite severe dealumination. We further characterized the Pd species to prove this inference.

### 2.3. The Analysis of Pd Species in Pd/Beta Zeolites

TEM measurement was first used to observe the particle size and distribution of Pd species in Pd/Beta zeolites with the result shown in Figure 5. For Pd/Beta-Al, the Pd species were homogeneously and highly distributed due to the observation of only sporadic particles. The average size of the observed particle was only about 3.6 nm and minified with progressively hydrothermal aging, indicating the high dispersion of Pd species regardless of hydrothermal aging. Compared to Pd/Beta-Al zeolite, more particles were observed on the Pd/Beta-C and Pd/Beta-Si zeolites with the average size of about 4.5 nm and 6.3 nm, respectively. This suggest that increasing the Si/Al ratio led to the accumulation of Pd species. Differently, the Pd/Beta zeolites with a medium Si/Al ratio (Pd/Beta-C) showed a decrease in particle size after the hydrothermal aging treatment, indicating the dispersion of Pd species. However, obvious agglomeration of Pd species occurred on Pd/Beta-Si zeolites after hydrothermal aging treatment. The particle size even grew to about 12 nm and 21 nm for Pd/Beta-Si-750HTA and Pd/Beta-Si-800HTA samples, respectively, indicating the significant accumulation of Pd species. This can explain why the NO_x_ storage amounts of Pd/Beta-Al and Pd/Beta-C increased, while that of Pd/Beta-Si decreased after hydrothermal aging treatment.

Furthermore, the Pd species on Pd/Beta zeolites with different Si/Al ratios were identified by H_2_-TPR measurement (Figure 6). In the temperature range of 0–50 °C, a H_2_ consumption peak appeared, which was attributed to the reduction of PdO species [15,40,41]. A negative peak at about 70 °C was attributed to decomposition of Pd hydride species [31,42]. For the fresh Pd/Beta-Al, the peak of PdO at 9 °C was observed and disappeared after the hydrothermal aging treatment, which again indicated that PdO was highly dispersed after the hydrothermal aging treatment. For the Pd/Beta-C zeolite, the peak at 9 °C was stronger than that of Pd/Beta-Al, suggesting the presence of more PdO species. Hydrothermal aging treatment at 750 °C also decreased the amounts of PdO species, which was similar to Pd/Beta-Al zeolites. Moreover, after the hydrothermal aging treatment at 800 °C, the temperature of PdO reduction shifted to higher temperature, also indicating the redispersion of Pd species as well as increased stability of PdO species. For the fresh Pd/Beta-Si zeolites, on the other hand, a large amount of PdO species were observed. After hydrothermal aging treatments at 750 °C and 800 °C, although the peak representing PdO species also decreased, this rarely meant high dispersion of Pd species due to the observation of a large-size particle in Figure 5i. Instead, the PdO species were reduced during hydrothermal aging due to the weak nature of ion exchange sites, and some Pd^0^ species were formed as indicated by the observation of a Pd^0^ diffraction peak in Figure 2c.

The effect of the Si/Al ratio on the local structure and state of Pd species for Pd/Beta zeolites was further investigated by X-ray adsorption fine structure (XAFS) (Figure 7). Pd K-edge X-ray absorption near-edge spectra (XANES) and extended X-ray absorption fine structure (EXAFS) spectra of PdO and Pd foil are listed in Appendix A. For XANES, the profiles were almost coincident before and after hydrothermal aging for the Pd/Beta-Al and Pd/Beta-C zeolites and resembled the profiles of reference PdO, indicating that Pd species existed in the form of PdO clusters or Pd^2+^ [9]. The results of EXAFS (Figure 7b) further indicated that the primary Pd species were the Pd^2+^ species coordinated with the zeolite framework for Pd/Beta-Al zeolites due to the absence of second scattering peak of Pd-Pd at ~3.0 Å [29,43]. However, for the Pd/Beta-C, PdO species existed as seen by the appearance of the second scattering peak at ~3.0 Å. The XANES and EXAFS profiles of Pd/Beta-Si zeolite were strikingly distinctive from the other two zeolites. With progressively hydrothermal aging, the XANES profile was closer to that of metallic Pd reference. The dramatical increase of the peak at ~2.5 Å (metal Pd-Pd) and decrease of the peak at ~1.5 Å (Pd-O) in EXAFS profiles also indicated the transformation from PdO into metallic Pd species during hydrothermal aging for Pd/Beta-Si [44,45,46].

The ionic Pd^2+^ were also verified by the DFIFTS measurement using CO as a probe molecule, with the result shown in Appendix A. All adsorbents showed distinct peaks in the range of 2215–2097 cm^−1^, ascribed to CO adsorbed on the Pd^n+^ (*n* = 0–3) ions, which indicated that isolated Pd exits at the exchange sites of the zeolite [9,47]. The peaks in the range of 2215–2097 cm^−1^ decreased with the increase of the Si/Al ratio, which indicated that the exchanged Pd ions were decreasing with the increase of the Si/Al ratio. Moreover, the hydrothermally aged Pd/Beta-Al and Pd/Beta-C also showed more intensified peaks than the fresh ones. This demonstrated the exchanged Pd ions species raised after hydrothermal aging.

Additionally, the in situ DRIFTS measurement was conducted after NO + O_2_ adsorption as shown in Appendix A. The peaks at 1570–1700 cm^−1^ were attributed to polydentate nitrate species on the zeolite framework [38]. The peaks at 1869 cm^−1^ and 1827 cm^−1^ were related to Pd(II)-NO and Pd(II)(NO)(CO) species, respectively [38]. The Pd/Beta-Al zeolites showed the largest NO_x_ adsorption regardless of hydrothermal aging. However, the hydrothermally aged Pd/Beta-Si showed scarce NO_x_ adsorption despite only a little of NO_x_ adsorption on Pd(II) ions on the fresh one. This indicated that Pd ions easily accumulated and were reduced to metallic Pd on Beta-Si zeolite during hydrothermal aging, which is consistent with TEM and XANES results.

## 3. Materials and Methods

### 3.1. Preparation of PNA Materials

Commercial H-Beta with specific Si/Al ratios of 6, 25 and 260, which were determined by inductively coupled plasma-optical emission spectroscopy (OPTIMA 8300), were written as Beta-Al, Beta-C and Beta-Si, respectively.

Pd/Beta zeolites were obtained by common impregnation method using Palladium (II) nitrate dihydrate (Sigma) as the Pd precursor. The Pd loading of all the adsorbent was 1 wt.%. The definite preparation method is as follows. First, 0.25 g Palladium (II) nitrate dihydrate was put into 600 mL deionized water and stirred for 10 min at room temperature. Second, 10.00 g of Beta was added, and the solution continued to be stirred for 2 h at room temperature. Third, the solution was transferred to a rotary evaporation flask with temperature of 80 °C for drying. The obtained sample was placed in an oven and dried for 12 h at 70 °C. Finally, the sample was calcined in a muffle furnace at 600 °C for 5 h with a heating rate of 5 °C/min. A synthetic path description diagram is shown in Appendix A.

In order to explore the hydrothermal stability of the adsorbent, the prepared adsorbent was hydrothermally aged at 750 °C and 800 °C for 16 h in 10% H_2_O/air, denoted as Pd/Beta-750HTA and Pd/Beta-800HTA, respectively.

### 3.2. Characterization of PNA Materials

The test of NO_x_ adsorption and storage performance on Pd/Beta zeolites was carried out on a self-built fixed reaction bed under a simulate exhaust condition. Mass flow controllers were used to control the flow rate of gas and the temperature is controlled by a K-type thermocouple. The Pd/Beta zeolite particle of 150 mg with 40–60 mesh was placed in a quartz tube (inner diameter of the quartz tube is 4 mm), and both ends of the Pd/Beta were blocked by quartz wool, in which case the sample was not blown into the detector. Pd/Beta were pretreated at 500 °C for 60 min with the feed atmosphere of 5% H_2_O, 10% O_2_/N_2_, following by dropping the temperature to 120 °C to collect background before testing. After deducting the background, 210 ppm NO_x_ (200 ppm NO, ~10 ppm NO_2_) and 200 ppm CO were introduced for isothermal adsorption at 120 °C. In order to understand more clearly the desorption ability of the adsorbent, we turned off NO_x_ and purged for 1 h at 120 °C before desorption. Finally, the temperature was raised to 520 °C with a heating rate of 10 °C/min for desorption process and the feed atmosphere was 5% H_2_O, 10% O_2_/N_2_. The Antaris™ IGS analyzer (Thermofisher Scientific, Waltham, MA, USA) was used to measure the concentration of the gas during the reaction.

X-ray diffraction (XRD) measurement was used to understand the crystallinity and structure of the zeolite. It was carried out in a Bruker D8 advance diffractometer (Bruker Corporation, Karlsruhe, Germany) with Cu kα as the light source. The scanning range was 4–45° and the step length was 0.02°.

A field emission transmission electron microscope (FETEM) (JEOL, Tokyo, Japan) was used to understand the morphology and the dispersion of Pd of the adsorbent. For analysis, the catalyst was dispersed over the channel and photographed with a camera.

Temperature-programmed reduction of hydrogen (H_2_-TPR) was used to understand the Pd species of the adsorbent. The experiment was carried out on a Micromeritics AutoChem 2920 chemisorption instrument (Micromeritics, GA, USA). The 50 mg adsorbent was placed into a U-tube and pretreated in O_2_/Ar atmosphere for 60 min at 500 °C. The temperature was reduced to −50 °C by liquid nitrogen injection followed by sweeping with Ar for 30 min. The temperature was controlled by programmed heating. Then, 10% H_2_/Ar was introduced into the atmosphere after the baseline was stable. The Pd species of the adsorbent were determined by recording the TCD signal.

Diffuse reflection infrared Fourier transform spectroscopy (DRIFTS) with CO as a probe was used to understand the Pd species, which was carried out on a Nicolet IS50 Fourier infrared spectrometer (Thermofisher Scientific, Waltham, MA, USA) equipped with an MCT detector. The adsorbent, ground into fine powder, was put into the sample pool and heated to 500 °C through the heating module at a heating rate of 10 °C/min. After Pd/Beta zeolite was pretreated at 500 °C for 60 min in the air inlet atmosphere of 10% O_2_ and N_2_, the adsorbent was cooled to room temperature. It stayed at room temperature for 50 min to collect background data. After deducting the corresponding temperature background, 200 ppm CO was injected, and the surface species were recorded at this time.

Pd K-edge X-ray absorption fine structure (XAFS) was carried out on beamline BL14W1 at Shanghai synchrotron radiation facility using XAFS beams Si (111) monochromatic light mirrors. Pd foil was used as the energy calibration, and PdO was used as the standard sample. The X-ray absorption near-edge structure (XANES) data were background-corrected and normalized using the Athena module implemented in the IFFEFIT software package [48].

^27^Al-nuclear magnetic resonance (^27^Al NMR) and ^29^Si-nuclear magnetic resonance (^29^Si NMR) values were collected on a nuclear magnetic resonance spectrometer (JEOL, Tokyo, Japan) equipped with a 3.2 mm MAS probe at a mass frequency of 12 KHz, which aimed to determine the framework Si and Al in the zeolite. The sampling times were 128 and 700 scans, respectively, and the relaxation delay was 5 s.

X-ray photoelectron spectroscopy (XPS) spectra were collected from the XPS spectrometer (Thermofisher Scientific, Waltham, MA, USA) to determine the valence states of Pd and the proportion of each valence state.

Diffuse reflection infrared Fourier transform spectroscopy (DRIFTS) was used to understand the NO_x_ adsorption and storage mechanism of adsorbent, which was carried out on a Fourier infrared spectrometer (Thermofisher Scientific, Waltham, MA, USA) equipped with an MCT detector. Pd/Beta zeolite was pretreated at 500 °C for 40 min in the atmosphere of 10% O_2_ and N_2_ followed by reducing the temperature to 150 °C. After cooling down, it stayed at 150 °C for 50 min to collect background data. After deducting the background, the isothermal adsorption was performed at 150 °C with the feed atmosphere of 500 ppm NO, 200 ppm CO and 10% O_2_ and N_2_.

## 4. Conclusions

Pd/Beta-Al has excellent NO_x_ adsorption and desorption capacity, suitable desorption temperature and perfect hydrothermal stability, which shows that it has great potential as a PNA candidate. Increasing the Si/Al ratio contributed to the formation of PdO, which led to the decrease of NO_x_ adsorption and storage capacity of fresh Pd/Beta zeolites. The desorption temperature also increases with the increase of the Si/Al ratio regardless of hydrothermal aging. After hydrothermal aging treatment, however, the NO_x_ adsorption–desorption capacity of Pd/Beta-Al and Pd/Beta-C significantly increased, which resulted from the redispersion of Pd species. The strong interaction with Pd ions and coordinated Al sites can survive from dealumination of Al-rich Beta zeolite. Hydrothermal aging treatment led to the formation of metal Pd^0^ species for Pd/Beta-Si zeolite, which made it an inappropriate candidate as a PNA material. The results inspired us to use Pd/Beta with a low Si/Al ratio as a PNA material candidate.

## Figures and Tables

**Figure 1 molecules-28-03501-f001:**
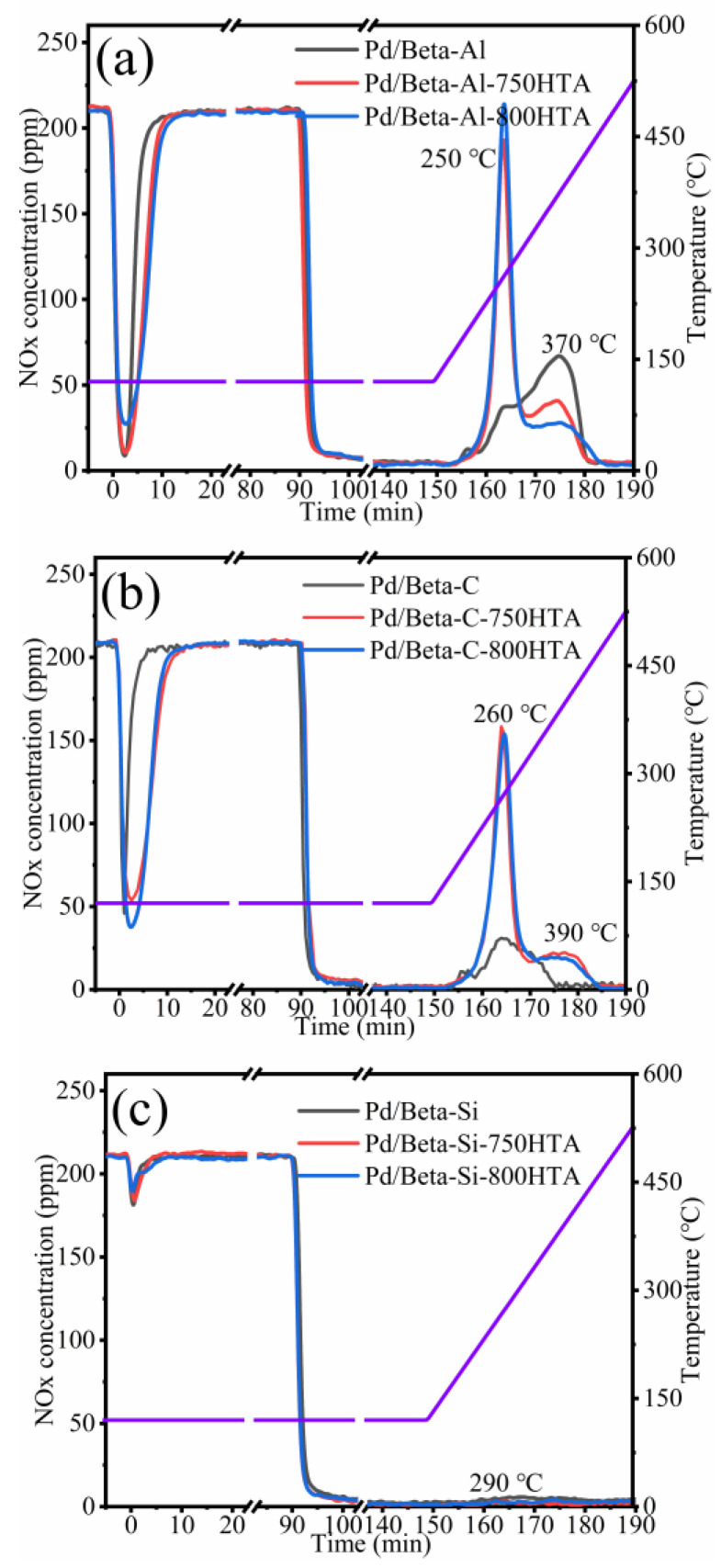
The NO_x_ adsorption and storage capacity of (**a**) Pd/Beta-Al, Pd/Beta-Al-750HTA and Pd/Beta-Al-800HTA; (**b**) Pd/Beta-C, Pd/Beta-C-750HTA and Pd/Beta-C-800HTA; (**c**) Pd/Beta-Si, Pd/Beta-Si-750HTA and Pd/Beta-Si-800HTA. Feed: 200 ppm NO; 200 ppm CO, 5%H_2_O, 10%O_2_ and N_2_ balance.

**Figure 2 molecules-28-03501-f002:**
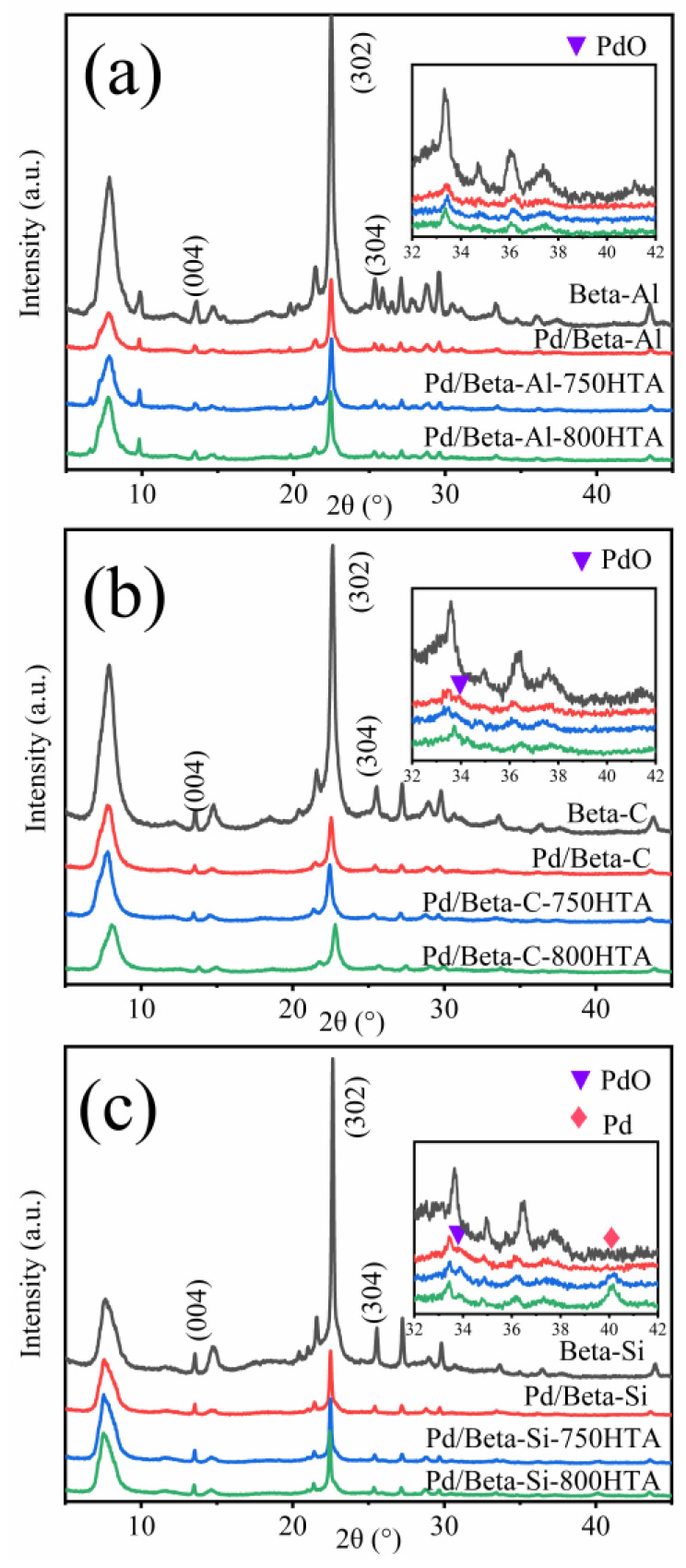
XRD patterns of (**a**) Beta-Al, Pd/Beta-Al, Pd/Beta-Al-750HTA and Pd/Beta-Al-800HTA; (**b**) Beta-C, Pd/Beta-C, Pd/Beta-C-750HTA and Pd/Beta-C-800HTA; and (**c**) Beta-Si, Pd/Beta-Si, Pd/Beta-Si-750HTA and Pd/Beta-Si-800HTA.

**Figure 3 molecules-28-03501-f003:**
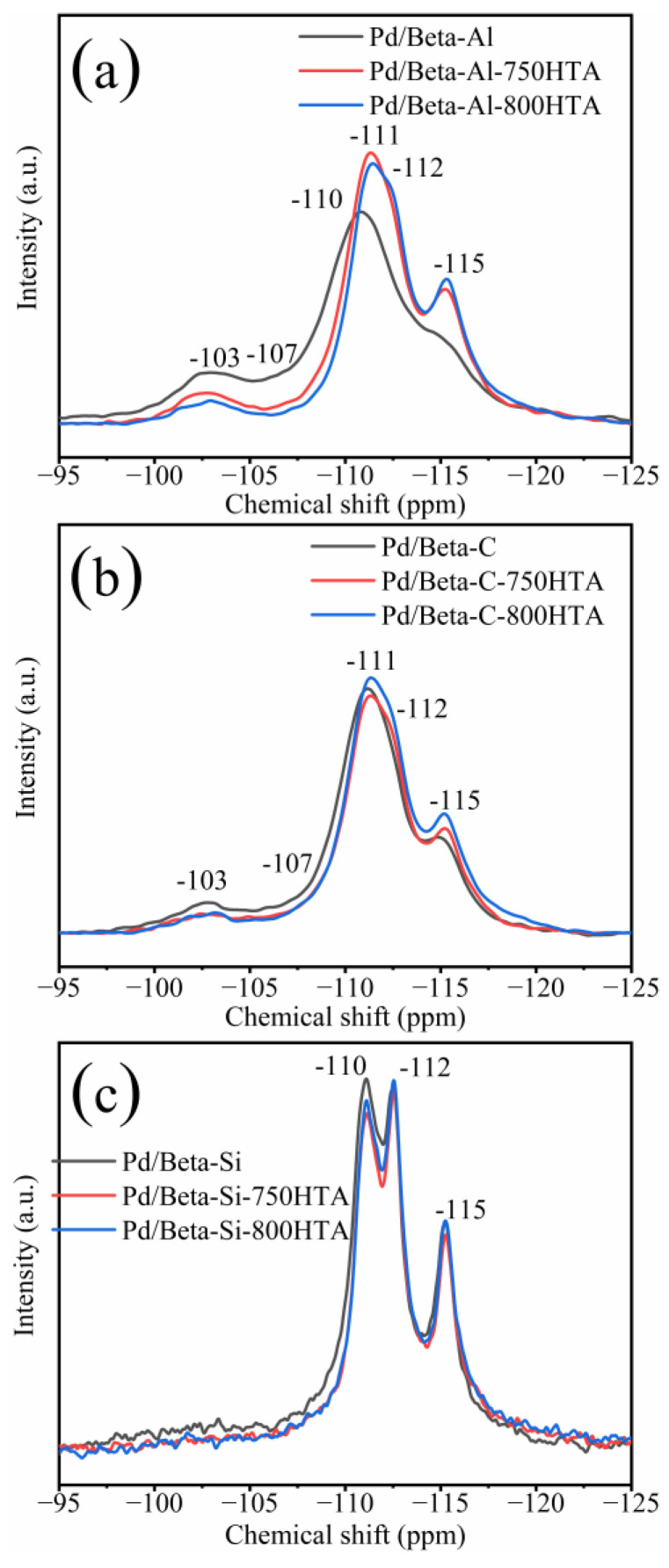
^29^Si NMR spectra for the (**a**) Pd/Beta-Al, Pd/Beta-Al-750HTA and Pd/Beta-Al-800HTA; (**b**) Pd/Beta-C, Pd/Beta-C-750HTA and Pd/Beta-C-800HTA; and (**c**) Pd/Beta-Si, Pd/Beta-Si-750HTA and Pd/Beta-Si-800HTA.

**Figure 4 molecules-28-03501-f004:**
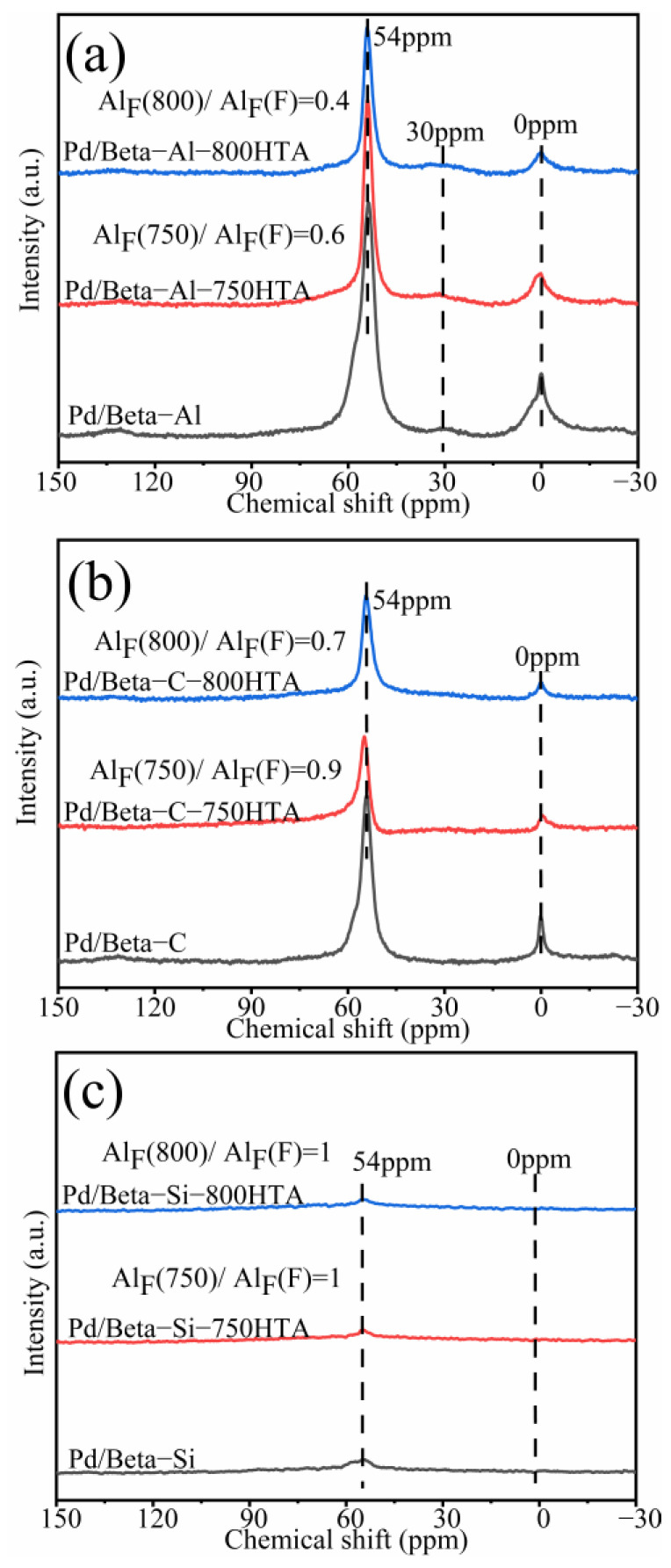
^27^Al NMR spectra for the (**a**) Pd/Beta-Al, Pd/Beta-Al-750HTA and Pd/Beta-Al-800HTA; (**b**) Pd/Beta-C, Pd/Beta-C-750HTA and Pd/Beta-C-800HTA; (**c**) Pd/Beta-Si, Pd/Beta-Si-750HTA and Pd/Beta-Si-800HTA.

**Figure 5 molecules-28-03501-f005:**
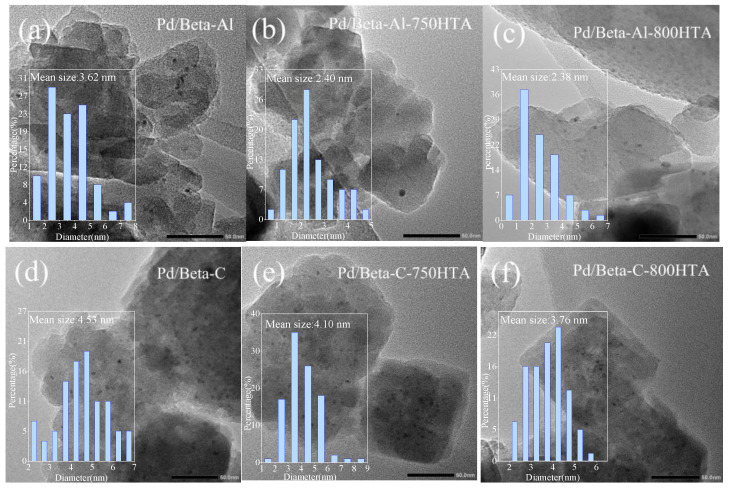
TEM images of (**a**) Pd/Beta-Al, (**b**) Pd/Beta-Al-750HTA, (**c**) Pd/Beta-Al-800 HTA, (**d**) Pd/Beta-C, (**e**) Pd/Beta-C-750HTA, (**f**) Pd/Beta-C-800 HTA, (**g**) Pd/Beta-Si, (**h**) Pd/Beta-Si-750HTA and (**i**) Pd/Beta-Si-800 HTA.

**Figure 6 molecules-28-03501-f006:**
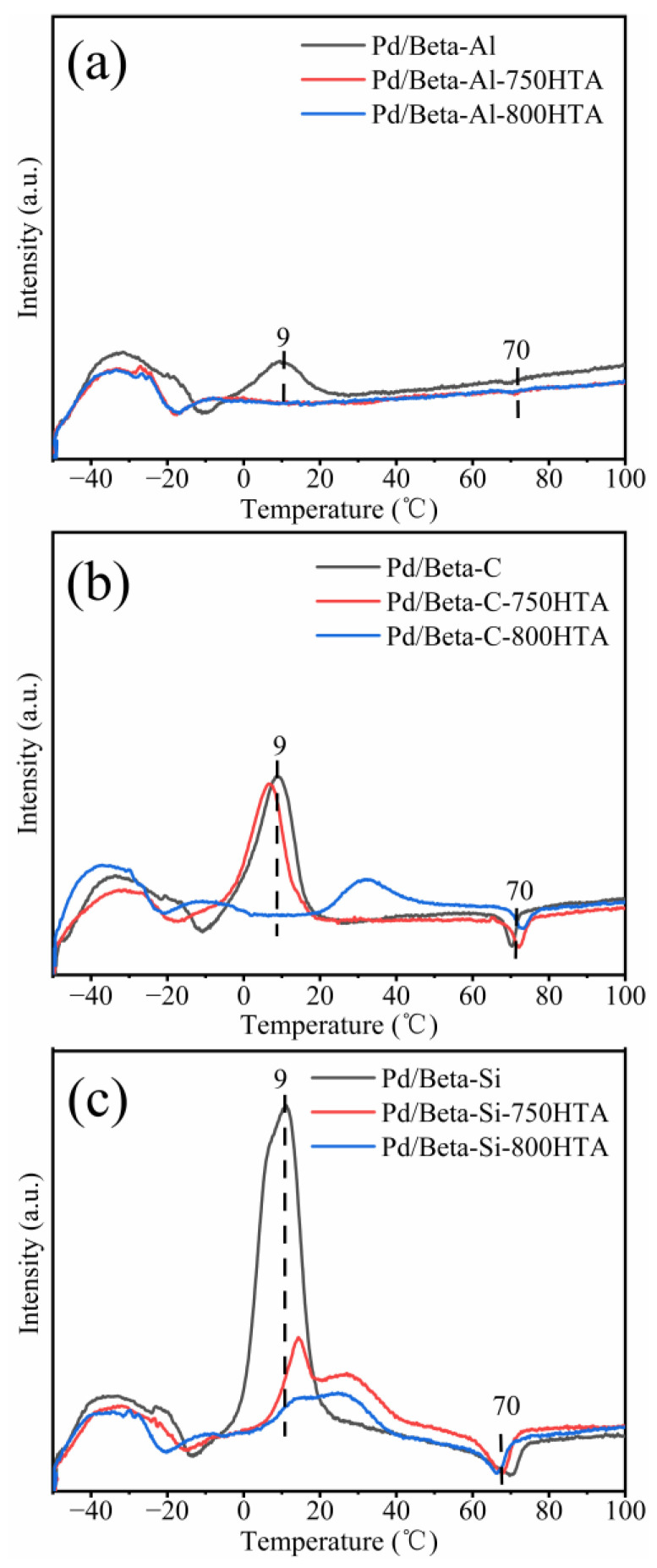
H_2_-TPR profiles of (**a**) Pd/Beta-Al, Pd/Beta-Al-750HTA and Pd/Beta-Al-800HTA; (**b**) Pd/Beta-C, Pd/Beta-C-750HTA and Pd/Beta-C-800HTA; (**c**) Pd/Beta-Si, Pd/Beta-Si-750HTA andPd/Beta-Si-800HTA HTA.

**Figure 7 molecules-28-03501-f007:**
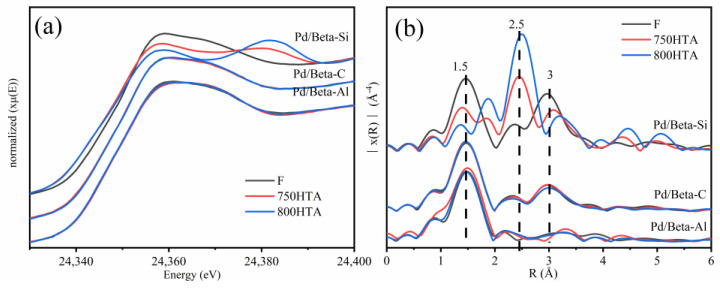
(**a**) Pd K-edge XANES spectra and (**b**) Fourier transforms of the k3-weighted Pd K-edge EXAFS of Pd1/Beta.

**Table 1 molecules-28-03501-t001:** The calculation of NO_x_ adsorption/desorption capacity of Pd/Beta with different Si/Al ratios Feed: 200 ppm NO; 200 ppm CO, 5%H_2_O, 10%O_2_ and N_2_ balance.

	Adsorption Capacity (umol/g)			Storage Capacity (umol/g)			NO_x_/Pd ^1^		
	F	750 HTA	800 HTA	F	750 HTA	800 HTA	F	750 HTA	800 HTA
Pd/Beta-Al	65.8	95.4	99.2	65.0	86.4	89.8	0.69	0.92	0.96
Pd/Beta-C	29.6	80.6	79.7	25.6	70.5	76.9	0.27	0.75	0.82
Pd/Beta-Si	4.6	4.5	4.7	2.9	0	0	0.03	0	0

^1^ The ratio of the NO_x_ storage amount and Pd loading.

## Data Availability

The data that support the findings of this study are available from the corresponding author upon reasonable request.

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
