# Peer review of "Effects of Si/Al Ratio on Passive NOx Adsorption Performance over Pd/Beta Zeolites"

_molecules, 2023, doi:10.3390/molecules28083501_

Round 1
Reviewer 1 Report
The main purpose of the submitted manuscript is to investigate the effect of Si/Al ratio on the NOx adsorption and storage capacity over Pd/Beta. The authors conclude that the NOx adsorption and storage capacity on Pd/Beta zeolites gradually decreased with the increase of Si/Al ratio.
The paper is well written, but there are some minor weaknesses that I feel must be addressed before the paper becomes acceptable.
Here some minor comments:
Section preparation of PNA materials should be rewritten. More detailes should be given. Can we add a synthetic path description diagram to briefly describe the synthesis method?
Characterization of PNA materials shoud contain information about the
detailed analysis of XANES measurement.
How does impregnation with Palladium (II) nitrate dihydrate affect the acidity of the system?.
Reviewer 2 Report
In the manuscript the effects on Si/Al ratio on the NOx adsorption and stability of Pd/Beta zeolite was investigated. Problem is rather topical. It was show that decrease of Si/f Al at zeolite lead to increase of NOx adsorption and store capacity. The article provides a sufficient number of modern research methods. The English of the paper is well. The article can be accepted for publication with minimal corrections. At the same time the several of comments should be noted:
1. The figures in the article look small and are not very informative. It is better to place them vertically in a row and increase the size
2. The reference consists of 47 citation and looks like almost all papers for this problem from South West regions. It will be better added more modern paper from European, American and Russian scientific teams.
Reviewer 3 Report
The subject of presented study is of great practical importance and it has been broadly investigated. The Authors quote almost 50 publications. The passive NOx adsorption from the diesel exhaust is an indispensible step before the catalytic denitrification. The adsorbents based on zeolites modified with Pt are recently very often investigated and applied in practice. Various types of zeolites (i.e either with low or high framework Si/Al, narrow and broad pores). The presented study concerns the correlation between Si/Al in zeolites Beta and adsorption efficiency of the Pd modified samples. This problem was already presented in literature, although the conclusions were not always quite uniform. The Authors used three samples of zeolite H-Beta with various Si/Al and impregnated them with the same content of Pd(NO3)2. The resulted samples were characterized with many respective analytical methods (XRD, NMR, XPS, XAFS, CO-DRIFT, TEM, H2-TPR) and their passive NOx adsorption efficiency was examined. The presented conclusions are similar to the most frequently published opinions that the Pd supported on low siliceous zeolites are more efficient passive adsorbents than those based on zeolites with high Si/Al. However, the latter show higher stability.
Regardless many modern techniques applied, the Authors do not indicate any new details regarding the nature of centers responsible for trapping the nitrogen oxides neither their mutual interactions. Perhaps, the IR or UV spectra could bring some helpful data. The XRD results (Fig. 2) are not very clear. May be, the patterns of pristine zeolites H-Beta could be useful to distinguish the Pd species.
The composition of the manuscript is not conventional. The Authors started the Results with the adsorption data, before the preparation and characterization of the samples under study. The English style also require some improvement (e.g. line 20: ,,Si/Al~260) has rarely activity for NOx “).
Summing up, I hope the manuscript could be reconsidered after major revision.
Round 2
Reviewer 3 Report
X/C/V